# Availability of Healthy Food and Beverages in Hospital Outlets and Interventions in the UK and USA to Improve the Hospital Food Environment: A Systematic Narrative Literature Review

**DOI:** 10.3390/nu14081566

**Published:** 2022-04-09

**Authors:** Sarah Richardson, Lorraine McSweeney, Suzanne Spence

**Affiliations:** Human Nutrition Research Centre, Population Health Sciences Institute, Faculty of Medical Sciences, M1.151 William Leech Building, Newcastle University, Framlington Place, Newcastle NE2 4HH, UK; sarah.richardson2442@outlook.com (S.R.); suzanne.spence@ncl.ac.uk (S.S.)

**Keywords:** food environment, healthy diet, hospital, systematic review, narrative synthesis

## Abstract

The aims of this systematic review are to determine the availability of healthy food and beverages in hospitals and identify interventions that positively influence the hospital food environment, thereby improving the dietary intake of employees and visitors. Embase, Medline, APA PsycInfo, Scopus, Google Scholar and Google were used to identify publications. Publications relating to the wider hospital food environment in the UK and USA were considered eligible, while those regarding food available to in-patients were excluded. Eligible publications (*n* = 40) were explored using a narrative synthesis. Risk of bias and research quality were assessed using the Quality Criteria Checklist for Primary Research. Although limited by the heterogeneity of study designs, this review concludes that the overall quality of hospital food environments varies. Educational, labelling, financial and choice architecture interventions were shown to improve the hospital food environment and/or dietary intake of consumers. Implementing pre-existing initiatives improved food environments, but multi-component interventions had some undesirable effects, such as reduced fruit and vegetable intake.

## 1. Introduction

Overweight and obesity are extremely prevalent across the UK and USA. In 2018, it was estimated that 67% of men and 60% of women in the UK had overweight or obesity [1], along with 71.6% of American adults in 2015/2016 [2]. A high body mass index (BMI) is linked to a range of non-communicable diseases, such as hypertension, type 2 diabetes and coronary heart disease [3], which has led to a significant number of hospital admissions associated with weight-related disorders. Between 2014 and 2015, it was estimated that the economic cost of overweight and obesity-related health complications to the National Health Service (NHS) was GBP 6.1 billion [4], while the healthcare cost of obesity in America was approximately USD 149.4 billion [5].

In addition to the high prevalence of obesity and overweight in the general population, healthcare employees demonstrate similar weight-management issues. One study carried out by Kyle et al. (2017) used data from the 2008–2012 Health Survey for England and found that 25.1% of the nurses surveyed had a BMI of 30 kg/m^2^ or higher, classifying them as ‘obese’. Furthermore, 32% of unregistered care workers, 26% of non-health-related NHS employees and 12% of other healthcare professionals also had a BMI of 30 kg/m^2^ or higher. These values are similar among American hospital staff members, with 27% of American nurses estimated to be obese [6].

A key cause of obesity is eating an excess of unhealthy foods. In the UK, the Office for Health Improvement and Disparities advises infrequent consumption of foods high in fat, salt and sugar [7]. American guidelines reflect the same general recommendations; according to the 2015–2020 Dietary Guidelines for Americans, a healthy diet should involve the restriction of saturated and trans fats, added sugars and salt [8]. Therefore, unhealthy foods can be defined as products that are high in these substances.

The range of healthy or unhealthy food and beverages available, food marketing techniques and the cost of food items in a specified setting can be referred to as the food environment [9]; this has a significant impact on the nutritional quality of food consumed by the general public. Studies have shown that there is an association between easier access to fast food and greater BMI and odds of obesity [10], suggesting that the food environment has a strong influence on weight status.

The general public has an expectation of hospitals and other healthcare environments to promote healthy behaviours, with 97% of participants in one survey indicating that hospitals should act as positive role models for healthy lifestyle behaviours [11]. Despite this, unhealthy foods are often found in hospital food outlets.

The aim of this review is to explore the extent to which healthy food and drink options are available to employees and visitors in hospital food environments and to determine which interventions are effective in reducing the purchase and consumption of unhealthy foods and beverages. This research may identify interventions that can improve the health and wellbeing of hospital employees and visitors, potentially leading to policy change to ensure healthy food is predominant in the wider hospital food environment.

## 2. Materials and Methods

The protocol associated with this systematic review was registered with PROSPERO, an international database of prospectively registered systematic reviews (reference number CRD42021223249). An amendment was made on 12 August 2021, detailing the repetition of database searching and a revised quality assessment method.

To find literature relevant to the current hospital food environment and interventions to improve the nutritional quality of food available to employees and visitors, a systematic search was carried out using five electronic databases: Embase, Medline, APA PsycInfo (all accessed via Ovid), Scopus and Google Scholar. Google was also utilised to ascertain suitable grey literature. Initial searches were carried out on 23 October 2020 and repeated on 21 July 2021 to detect new publications.

Suitable keyword search terms were identified; controlled search terms included “hospital”, “convenience food”, “healthy diet”, “automatic food dispensers” and “nutritional value”. Key words were amended slightly for each database; full search terms are listed in Appendix A.

Eligibility criteria were established to aid the selection of relevant publications. Some criteria were used to narrow the scope of the research to facilitate a detailed review of source material within the timeframe available. Two researchers carried out independent eligibility screening using Rayyan [12], and disputes were resolved via discussion with a third researcher.

Publications were included in the present review if they related to the wider hospital food environment (i.e., food outlets accessible to hospital employees and/or visitors) in the UK or the USA. No restrictions were placed on publication dates. Publications were excluded if they focused on food available only to patients in hospital wards. Additional exclusion criteria included studies with no full-text sources available, studies written in a language other than English and studies which involved systematic reviews or meta-analyses.

Search results were imported to Endnote [13]; duplicates were removed before the sample was exported to Rayyan [12]. Studies were initially screened based on the adherence of their titles and abstracts to the eligibility criteria. Included studies were further refined by screening full texts and removing ineligible records.

Several key pieces of data were extracted from each study in a standardised template by one researcher. Extracted information included author names, year of publication, country, study design, aim, duration, intervention/observation methods, outcome measures and results. A quality assessment and risk of bias analysis was also carried out on each source by one researcher using the Quality Criteria Checklist for Primary Research from the Academy of Nutrition and Dietetics [14]. See Appendix B, Table A1 for the full data extraction table.

The key outcomes of interest were the nutritional quality of food and beverages currently available to employees and visitors in hospitals as well as interventions aiming to improve the nutritional quality of products, awareness of nutritional values, dietary intake or overall health of hospital employees and visitors. Summary measures for these outcomes varied greatly between eligible publications. Due to the heterogeneity of summary measures and study designs, a quantitative synthesis or meta-analysis was not possible; consequently, a narrative synthesis was undertaken.

Studies were initially grouped into categories based on study type (i.e., observations and interventions). Interventions were allocated to sub-categories to allow for a well-structured narrative synthesis. Sub-categories included educational, labelling, financial, choice architecture, pre-existing guideline implementation and multi-component interventions.

## 3. Results

Of the 806 search results initially identified from databases and search engines, 40 studies met the eligibility criteria. The most common reason for exclusion was the irrelevance of study outcomes to the research question. This was often due to a focus on patient meals rather than the food available to hospital employees and visitors. A PRISMA flow diagram [15] displays the inclusion and exclusion process (Figure 1).

### 3.1. Participant Characteristics

Fifteen publications reported the number of hospital employees, students or visitors involved in observations or interventions [16,17,18,19,20,21,22,23,24,25,26,27,28,29,30], while twelve reported the number of food outlets [31,32,33,34,35,36,37,38,39,40,41,42] and ten reported the number of healthcare facilities involved [43,44,45,46,47,48,49,50,51,52]. Three studies recorded the number of food outlets and survey respondents [53,54,55]. In total, the eligible publications reported the involvement of 18,171 participants, 139 food outlets and 529 hospitals and healthcare facilities.

### 3.2. Countries

In the UK, 5 interventions [31,34,38,39,40], 7 observations [17,23,35,37,44,45,51] and 1 mixed methods design [41] were reported; in the USA, 14 interventions [16,20,24,25,28,29,30,33,36,42,43,48,53,54,55], 10 observations [18,19,21,27,32,46,47,49,50,52] and 2 mixed methods studies [22,26] were reported.

### 3.3. Study Design

Observational studies (*n* = 17) employed a range of techniques, such as interviews, focus groups and cohort studies. Of the intervention studies (*n* = 20), 8 utilised a randomised controlled trial design [16,28,29,31,33,34,38,53], and 12 utilised quasi-experimental methods [20,24,25,30,36,39,40,42,43,48,54,55]; additionally, 3 studies employed mixed methods [22,26,41], incorporating a range of techniques, such as conducting interviews and collecting sales figures. The full data extraction can be seen in Table A1. According to the Quality Criteria Checklist for Primary Research, 11 of the eligible publications met high quality and risk of bias standards [16,18,26,27,28,29,31,38,48,49,53], while 29 were considered neither particularly strong nor particularly weak (Table A1) [17,19,20,21,22,23,24,25,30,32,33,34,35,36,37,39,40,41,42,43,44,45,46,47,50,51,52,54,55].

### 3.4. Observations

Five observational studies explored hospital food outlet adherence to pre-existing standards and guidelines [32,35,37,46,51]. Sustain (2017) found variation in the hospital food environments between 30 hospitals, with around 50% complying with standards listed in the NHS contract. Healthy options were also found to be more prevalent than unhealthy options in vending machines [51]. Similarly, James et al. (2017) investigated 30 food outlets across two NHS hospitals and their adherence to The National Institute for Health and Care Excellence (NICE) Quality Standard 94. Quality Standard 94 includes three quality statements relating to the availability of healthy options in vending machines (statement 1), nutritional information on menus (statement 2) and prominent display of healthy options (statement 3). Adherence to statements 1 and 2 was poor; only 10% of food products and 53% of drinks available in vending machines were classified as healthy and nutritional information was not available on menus at either hospital. Adherence to statement 3 was mixed, as both healthy and unhealthy options were displayed prominently in food outlets [35].

In 19 facilities across California, Lawrence et al. (2009) found that 81% of food in vending machines did not adhere to the California state nutrition standards for schools. Carbonated drinks were the most common beverages in vending machines, with advertisements for these beverages being prevalent. At the time of the study, 60% of facilities had already adopted or were beginning to implement nutritional standards for vending machines [46].

Mohinra et al. (2021) investigated the food environment in a dental hospital and found that beverages met Commissioning for Quality and Innovation (CQUIN) targets for sugar content; however, foods high in fat, salt and sugar were displayed in prominent locations, and unhealthy options were more affordable than healthy options [37].

Derrick et al. (2015) assessed the nutritional quality of cafeteria meals in relation to LiVe Well Plate guideline adherence. On average, food outlets that adhered to the LiVe Well Plate guideline had significantly higher nutrition composite scores than those which did not, particularly for point-of-purchase options, suggesting healthier food environments [32].

The diverse nutritional quality of cafeteria meals was also reported by Jaworowska et al. (2018). Variation was identified between different meals in the same outlet and between the same meals at different facilities; the majority of meals were high in saturated fat, while 69% of meat-based dishes and 43% of vegetarian dishes were high in salt content [44].

Findings were similar within paediatric hospitals or clinics. Across 14 facilities, Lesser et al. (2012) reported that most food outlets offered healthy options and half displayed nutritional information at the point of purchase. However, the majority had high-calorie options positioned close to point-of-purchase and promoting unhealthy options on signs was more common than promoting healthy options. Furthermore, half of the cafeterias had no healthy hot meals [47]. In vending machines accessible to children, Kibblewhite et al. (2010) found that none of the food-based or mixed food and drink vending machines contained 50% or more healthy food options. Meanwhile, 13% of drinks machines in paediatric clinics and 9% of drinks machines in other areas of the hospital contained 50% or more healthy options. Advertisements for brands associated with unhealthy products were also commonly found on vending machines [45].

Parental visitors to a paediatric hospital were broadly dissatisfied with the food environment. Food options were considered restrictive, and concerns were raised regarding the quality, freshness and positioning of products. Participants also felt that food available in the hospital food environment contradicted healthy eating messaging on signage [23].

Interviews, focus groups and surveys were also carried out with hospital employees and non-parental visitors. Bak et al. (2020) reported that nursing students believed that few healthy food options were available in hospitals. The students indicated that subsidising healthy foods could improve the hospital food environment and positively influence eating behaviours among nurses [17]. Barriers and facilitators to healthy eating were identified via interviews with 17 food service managers, carried out by Lederer et al. (2014). Only four of the respondents reported that their cafeteria followed nutrition standards set by the hospital (*n* = 3) or by the American Heart Association and the Academy of Nutrition and Dietetics (*n* = 1). The majority of respondents said that consumer-related factors, such as customer satisfaction and demand, were barriers to healthy food implementation [21]. Consumer satisfaction was also cited as a potential barrier by food service managers in a study by Jilcott Pitts et al. (2016). Other challenges included profit implications and training costs, while potential facilitators of healthy eating included altered positioning of healthy and unhealthy options and signage promoting healthy options [19]. Furthermore, Liebert et al. (2013) also highlighted profits and resources as potential barriers to implementing nutrition interventions. Despite this, over 80% of respondents were concerned about eating well and stated that they would be more likely to do so if healthy options were cheaper than unhealthy options. Additionally, 73% were in favour of the taxation and subsidisation of products based on nutritional content [22].

The impact of hospital location or average visitor socioeconomic status on the hospital food environment was also explored. Winston et al. (2013) found no significant relationship between the socioeconomic status of the local area and the nutrition composite score of the hospital [52]. In contrast, a study by Goldstein et al. (2014) found that physicians seeing mostly patients of higher socioeconomic status were more likely to report high levels of nutritional support compared to those seeing patients of lower socioeconomic status [18].

### 3.5. Interventions

Hospital food environment interventions can be grouped into several categories. Studies incorporating interventions from three or more categories are classified as “multi-component”; these studies are grouped and explored separately under the “Multi-Component Interventions” subheading.

#### 3.5.1. Educational

Educational interventions aim to increase consumer knowledge of the nutritional guidelines or the nutritional content of foods. Of the seven educational interventions, four utilised signage or flyers to increase awareness of the nutritional content of products.

Allan and Powel (2020) assessed the impact of point-of-purchase signage on the nutritional quality of purchases and found that signage reduced the calorie content of purchases, reduced sugar content in some circumstances and had no impact on fat content [31]. Webb et al. (2011) also introduced nutritional labelling on posters or nutritional labelling on posters and a point-of-purchase menu board. More consumers noticed the nutritional information when posters and menu boards were used, compared to only posters. Nutritional labelling on posters and menu boards was also associated with the increased purchase of lower-calorie snacks and side dishes but made no significant difference in the nutritional content of entrée purchases [55].

When combining signage with traffic light labelling, Sonnenberg et al. (2013) found that nutritional content, taste and price became more important to consumers, while convenience became less important. Participants who were influenced by nutritional information bought more healthy products than those who were not [25]. Block et al. (2010) introduced signage and flyers about the health implications of regular soft-drink consumption, along with taxation. Education alone had no significant effect on sugar-sweetened soft drink sales. However, education enhanced the effects of taxation, reducing soft drink sales by 10% compared to price increases alone [53].

Two studies used digital methods to deliver nutritional information [16,28]. Abel et al. (2015) sent texts or emails to participants regarding calorie reference values. Participants who received the information were twice as likely to know reference values compared to a control group, but this did not appear to alter calorie consumption or portion sizes [16]. Thorndike et al. (2021) used emails (and letters) to provide feedback on food choices. The number of healthy purchases increased while the number of unhealthy purchases decreased. These effects remained significant at a 24-month follow-up, but there was no significant change in weight status [28].

Another study by Thorndike et al. (2016) used social norm feedback in combination with financial incentives. Social norm feedback alone led to a 1.8% increase in healthy purchases, but this was not statistically significant. Combining social norm feedback with a financial incentive resulted in a 2.2% increase in healthy purchases, and employees rated healthiest at baseline were influenced most greatly by the interventions [29].

#### 3.5.2. Labelling

Six studies explored the effects of labelling [25,27,30,33,42,54]. Elbel et al. (2013) added ‘less healthy’ labels to some items, and this increased purchase of healthier options by 7% [33]. Sato et al. (2013) added calorie, fat and sodium content information to packaging. A non-significant increase in healthier options sold was observed, along with a significant decrease in the total number of meals sold per day. Despite this, 71% of customers who noticed the intervention reacted positively to it, and 50% claimed that the labels influenced them to purchase a healthier option [54].

The most common form of labelling was traffic light labelling. Sonnenberg et al. (2013) investigated the impacts of traffic light labelling and nutritional signage on customer food-related attitudes. The intervention increased the importance of health and nutrition to participants, and more participants claimed to use nutritional information when making food choices during the intervention compared to pre-intervention. More healthy options were purchased by those who were influenced by the labels than those who were not [25].

Traffic light labelling reduced the number of unhealthy purchases in three studies. Whitt et al. (2018) assessed the impact of traffic light labelling on food choices compared to cartoon labelling. Traffic light labelling decreased unhealthy food purchases by 7% from baseline, while cartoon labelling increased the number of unhealthy purchases by 1% from baseline and by 5% from the washout period [42]. In two studies, Thorndike et al. (2014, 2019) found that traffic light labelling decreased the proportion of red-labelled products purchased [30], which resulted in fewer calories purchased and potential employee weight loss [27].

#### 3.5.3. Financial

The effects of financial interventions were primarily investigated via taxation and subsidisation of products. Elbel et al. (2013) found that taxing unhealthy products increased the proportion of healthy purchases by 11.5% from baseline and that this was associated with fewer unhealthy purchases and an increased proportion of healthy beverage purchases [33]. Similarly, Block et al. (2010) found that increasing the prices of sugar-sweetened soft drinks decreased sales by 26% [53]. Patsch et al. (2016) utilised taxation and subsidisation and found significant increases in the proportion of healthy alternatives sold along with decreases in the number of traditional, less healthy products sold [24].

In addition to feedback-based interventions, Thorndike et al. (2016) offered financial incentives for healthy purchases. Feedback plus a financial incentive led to a 2.2% increase in healthy purchases, compared to a 1.8% increase for feedback alone and a 0.1% increase for the control group [29]. In a second study by Thorndike et al. (2021), the intervention increased the purchase of healthy products by 7.3% and decreased the purchase of unhealthy options by 3.9%. This effect did not lead to significant weight loss in the intervention group [28].

#### 3.5.4. Choice Architecture

Choice architecture was implemented in several ways; the most prevalent method was altering the proportion of healthy and unhealthy products available to purchase. Three studies focused on products available in vending machines. Griffiths et al. (2020), Grivois-Shah et al. (2018) and Pechey et al. (2019) found an increase in the number of healthy products purchased [43], a decrease in the amount of calories purchased [34,38] and mixed results regarding the financial impact of the intervention [34,43]. Simpson et al. (2018) conducted a similar study in a hospital shop but found no significant difference in the relative proportion of healthy options sold between pre- and post-intervention sales data [40].

Thorndike et al. (2014) took a different approach and utilised choice architecture by making healthy options more visible. The impact of choice architecture itself is unknown, as it was only assessed in combination with traffic light labelling. Nevertheless, the overall intervention decreased the proportion of unhealthy product sales by 3% and increased the proportion of healthy product sales by 5% after two years [30].

Public Health England, as previously known (2018), investigated the impact of altering product positioning in vending machines; this is explored in more depth in the ‘Implementing Standards and Guidelines’ section [39].

#### 3.5.5. Implementing Standards and Guidelines

Three studies assessed the impacts of supporting the implementation of pre-existing standards for hospital food outlets [39,41,48]. Moran et al. (2016) encouraged the implementation of the Healthy Hospitals Food Initiative and improved adherence to the programme. The nutritional quality of the hospital food environment also improved [48].

Stead et al. (2020) focused on the implementation of Healthcare Retail Standards in hospital shops. Compliance with these standards had no effect on the number of fruit products available but decreased the number of chocolate-based options and the number of promotions for these products. The standards also reduced meal-deal sales [41].

Public Health England (2018) altered the content of vending machines in line with Government Buying Standards for Food and Catering. In drinks machines, these changes decreased calorie and sugar content of purchases and increased proportion of ‘diet’ beverages sold. In food machines, sales of crisps decreased while sales of confectionary and dried fruit and nut products increased [39].

#### 3.5.6. Multi-Component Interventions

Stites et al. (2015) carried out a study involving choice architecture, financial incentives and educational components. Hospital employees were taught mindfulness techniques, encouraged to pre-order meals and, for part of the intervention, provided with vouchers for the cafeteria. This intervention resulted in lower calorie and fat purchases compared to a control group, with and without the vouchers. Despite the dietary changes, weight loss was not significant among the intervention group [26].

LaCaille et al. (2016) incorporated signage, traffic light labelling and choice architecture into a nutrition intervention, alongside encouraging physical activity participation. The control group experienced a greater average reduction in waist circumference than the intervention group after six months (but not after 12 months). As well as this, the intervention group experienced a significant decrease in fruit, vegetable and fibre intake over the course of the study. Consumption of foods high in sugar and fat also decreased [20].

Mazza et al. (2017) also reported mixed results. Financial interventions and traffic light labelling were combined with health and social norm messaging and choice architecture. Financial interventions and traffic light labelling increased healthy beverage purchases by 2.9%. The addition of health and social norm messaging and grouping items into nutritional categories further increased healthy beverage purchases. Healthy crisp sales increased by 5.4% when traffic light labelling was introduced and by 6% when health messaging was implemented. However, healthy crisp sales decreased by 5.9% when the price of water was reduced, suggesting that this financial intervention nullified the beneficial effects of traffic light labelling [36].

## 4. Discussion

The observational studies carried out across the UK and USA suggest that the quality of the wider food environment is diverse. Compliance with pre-existing standards and guidelines is varied [32,35,37,46,51] and the nutritional quality of cafeteria meals differs between meals and facilities [44,47]. A lack of healthy food options was reported in vending machines, while the availability of healthy beverage options was slightly greater [45]. Hospital visitors and employees reported concerns regarding the quality, freshness and positioning of healthy and unhealthy options [23] and believed there was a lack of healthy options available [17]. Barriers to the implementation of healthy eating initiatives were also identified, including customer satisfaction [19,21] and profit implications [19,22], although participants were in favour of a financial intervention to encourage healthy food and beverage choices [22]. Findings relating to the impact of socioeconomic status on the hospital food environment are inconsistent [18,52].

Utilising signage and flyers is associated with the reduced calorie content of purchases [31], and displaying nutritional information on menu boards and posters increases the purchase of low-calorie options compared with using posters alone [55]. Digital methods of communicating nutritional information can increase knowledge of reference values [16] and increase purchase frequency of healthy options whilst decreasing purchase frequency of unhealthy options [28], but effects on calorie consumption are contested [16,28] and these interventions have no significant impact on weight-related outcomes [28]. Moreover, educational interventions can be successfully incorporated with traffic light labelling [25] and financial interventions [29,53].

Adding simple labels to products, marking them as ‘less healthy’ or giving some nutritional information, is associated with an increase in the number of healthy purchases [33] but also with decreased total purchases per day [54]. Nevertheless, labelling interventions are viewed positively by consumers [54]. Traffic light labelling has been shown to increase the importance of nutrition to participants [25], increase the number of healthy purchases [25], reduce the number of unhealthy purchases [30,42] and reduce the calorie content of purchases [27]. It was predicted that this could lead to consumer weight loss, provided that no other lifestyle alterations occurred [27].

Taxation on unhealthy products or subsidisation of healthy products was found to be associated with an increased proportion of healthy purchases [24,33] and a decreased number of sugar-sweetened soft drink purchases [53]. Financial incentives were also found to effectively increase healthy purchases and decrease unhealthy purchases [28,29], but no impact on weight-related outcomes was identified [28].

Altering the proportion of healthy options available to purchase from vending machines was found to increase healthy purchases [43] and decrease the calorific content of purchases [34,38]. This type of intervention may have undesirable financial outcomes for food outlets, but this remains unclear [34,43].

One study found no change in the proportion of healthy options sold before and after a choice architecture intervention in a hospital shop [40]. However, another study reported that displaying healthy options more prominently reduced unhealthy purchases and increased healthy purchases when combined with traffic light labelling [30]. Choice architecture interventions also increased ‘diet’ beverage sales and reduced the total sugar content of purchases [39].

Encouraging implementation of pre-existing standards and guidelines is associated with an overall improvement in the hospital food environment [48] and decreased availability of unhealthy products [41]. In beverage vending machines, implementing governmental standards increased the proportion of ‘diet’ beverage sales and reduced the sugar and calorie content of purchases [39]. However, adherence to these standards has also been shown to reduce sales of meal deals [41] and increase sales of confectionary [39].

Multi-component interventions have been carried out with a range of study designs. These interventions have been shown to reduce calorie and fat content of purchases [26], reduce consumption of foods high in sugar and fat [20], increase healthy beverage purchases [36] and increase the purchase of healthy snack options [36]. However, certain multi-component interventions have also resulted in decreased fruit, vegetable and fibre intake [20] and decreased sales of healthy snack options [36]. These interventions were not associated with significant weight loss [20,26].

This review has several strengths. Study screening was independently conducted by two researchers, and the process was blinded to reduce the impact of researcher bias. Additionally, conducting a narrative synthesis allowed the integration of material that would have been incomparable using quantitative synthesis. Therefore, the heterogeneous data have been compiled into a useful summary to inform further research. However, the heterogeneity of study designs and outcome measures restricts the use of quantitative synthesis and meta-analysis to summarise findings. Additionally, many studies involve multiple interventions, making it difficult to determine the impact of each intervention on outcomes. This limits the strength of recommendations.

Another limitation of this review is our pragmatic decision to restrict the search to studies carried out in the UK and USA. This decision was taken because the number of studies carried out in the UK and USA allowed for an in-depth analysis of all relevant literature within the available time frame. It should be noted that the exclusion of studies from other countries limits the wider generalisability of findings.

Furthermore, some studies took place several years ago, meaning that information about the ‘current hospital food environment’ may no longer be valid. One observation found that 60% of facilities were beginning to adopt vending machine nutrition standards in 2009 [46], so vending machine nutritional quality could have since changed. Nevertheless, poor nutritional quality in vending machines was reported in 2017 [35], indicating that concerns about healthy vending machine options may remain relevant.

The majority of interventions discussed in this review can be used to improve nutrition awareness, eating behaviour or the overall hospital food environment. Studies that surveyed hospital employees or visitors on their acceptance of these interventions reported that over 70% of responses were positive [22,54].

Some studies show that combining multiple interventions can improve the nutritional quality of food purchases. However, multi-component interventions may have the potential to lead to detrimental impacts, such as reduced fruit and vegetable consumption and reduced sales of healthy snacks [20,36]. Consequently, more robust study designs are required to identify the most effective intervention combinations in multi-component studies.

Further interventions are needed in the UK to investigate the most effective methods of improving the nutritional quality of employee and visitor diets. Research into the associations between food environment and food intake in a variety of settings, other than hospitals, would also be valuable. Moreover, some interventions included in this review would benefit from being replicated to generate an evidence base with consistent outcome measures. This would produce more homogenous data, facilitating quantitative synthesis and meta-analysis. A more precise estimated effect size would be generated, thereby strengthening practice and policy recommendations.

Timely implementation of public health interventions, such as altering food environments and encouraging healthier diets, is especially pertinent in the wake of the COVID-19 pandemic. Dietary patterns high in fat, salt and sugar contribute to the prevalence of obesity and type II diabetes, which increase the risk of severe COVID-19 outcomes [56]. By altering the hospital food environment, healthy food and beverages could be made the easiest option to purchase, thereby improving dietary quality and potentially reducing the risk of ill-health among hospital employees and visitors. Food environment interventions could also reduce the discrepancy between health messaging and poor hospital food environments, ensuring that hospitals act as positive role models for healthy lifestyle behaviours.

## 5. Conclusions

In conclusion, the quality of the hospital food environment varies within and between facilities. Hospital visitors and employees are generally receptive to food environment interventions and a variety of designs can be used to improve the hospital food environment and increase the proportion of healthy purchases. However, multi-component interventions can have neutral or detrimental effects on participant eating behaviours depending on the design. Therefore, further research that also encompasses studies beyond the UK and USA is required to determine the most effective combinations within multi-component interventions.

## Figures and Tables

**Figure 1 nutrients-14-01566-f001:**
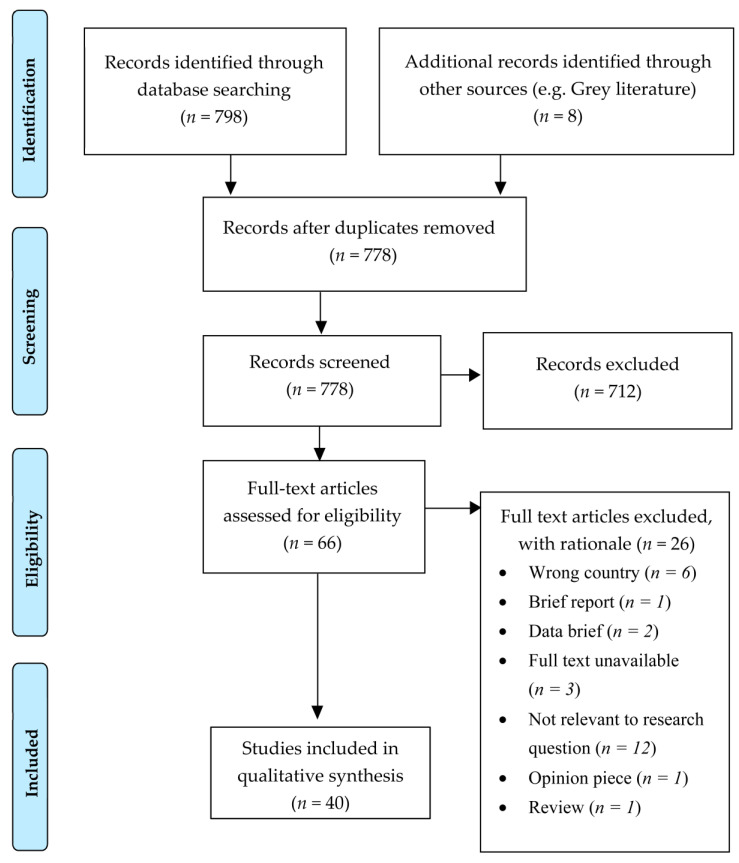
PRISMA 2009 flow diagram [15], detailing the number of studies included and excluded at each stage of the screening process.

## Data Availability

Not applicable.

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
