# Peer review of "Availability of Healthy Food and Beverages in Hospital Outlets and Interventions in the UK and USA to Improve the Hospital Food Environment: A Systematic Narrative Literature Review"

_nutrients, 2022, doi:10.3390/nu14081566_

Round 1
Reviewer 1 Report
Within the introduction the aim of the review based on a reported association between food environment and food consumption. To underline this association the authors named only one study exploring the association between fast food and BMI.
A more sustainable summary of the existing work regarding the association between food environment and food intake, e.g. in different work settings would improve the review and the readers understanding.
Author Response
Thank you for taking the time to review our manuscript. We really appreciate your feedback and agree that a broader piece of work regarding the association between food environment and food intake in different settings would be valuable.
However, our particular focus in this review was to consider the hospital setting as the key objective and the aims and environment were specified on lines 59-64.
To address this feedback, we have added a sentence to the discussion to acknowledge the value and importance of studying the relationship between food environment and food intake in a variety of settings, other than within hospitals (lines 445-447).
Reviewer 2 Report
This is a very interesting study, which addresses a topic that has not been studied in depth.
Among its strengths, I highlight how interesting it is to study an environment that is conducive to educating about healthy food choices as well as the wide variety of types of interventions described and included in the article.
I think it should be explicitly mentioned as a weakness of the study, that it was only focused on the work carried out in the United States and the United Kingdom, leaving aside any studies performed elsewhere in the world. Likewise, the title should also reflect this aspect, since it would seem that studies from all parts of the world are included, which is not the case.
Author Response
Thank you for taking the time to review our manuscript. We agree that our pragmatic decision to restrict the review to the UK and USA is a limitation. We have addressed this feedback by emphasising this limitation (lines 423 - 427) and have added the need for wider scope in our conclusions (lines 468-469). As recommended, we have also reflected the choice to restrict to the UK and USA in the title.